# Pharmacogenomics, CYP2D6, and Tamoxifen: A Survey of the Reasons Sustaining European Clinical Practice Paradigms

**DOI:** 10.3390/medicina55070344

**Published:** 2019-07-05

**Authors:** Sara S. Reis, Ana S. Carvalho, Rúben Fernandes

**Affiliations:** 1School of Engineering and CIETI, Porto Polytechnic (P. Porto), 4249-015 Porto, Portugal; 2Institute of Bioethics, Portuguese Catholic University, 4169- 005 Porto, Portugal; 3School of Health, Porto Polytechnic (P. Porto), 4200-072 Porto, Portugal; 4Institute for Research and Innovation in Health (i3S), Porto University, 4200-135 Porto, Portugal

**Keywords:** pharmacogenomics, clinical policies, clinical bias, breast cancer, tamoxifen, CYP2D6, hormone therapy, cultural sample control

## Abstract

Tamoxifen is a drug that is often used in the clinical management of breast cancer. CYP2D6 is a key metabolizing enzyme that is involved in the conversion of tamoxifen to its active drug metabolites. CYP2D6 has several alleles that metabolize tamoxifen and other drugs at different rates that can alter therapeutic impact, a characteristic that renders it one of the most studied enzymes in the field of pharmacogenetics. *Background and objectives*: Portugal has no implemented measures based on pharmacogenomics analysis prior to therapy that might function as a cultural sample control when analyzing the individual and economic factors present in clinical practice paradigms. Therefore, we aim to investigate the impact of CYP2D6 genotyping of the tamoxifen metabolizing enzymes in the clinical management of breast cancer patients. *Materials and Methods*: Qualitative/quantitative studies regarding the impact of pharmacogenomics in breast cancer; personal interviews in different Portuguese laboratories within hospital setting using a survey. Analysis of data through interviews to management board and/or decision makers from major oncological centers. *Results*: Reasons for common adoption of pharmacogenomics practice are contradictory and based both in economic factors and cultural/clinical bias. *Conclusions*: This research study identifies specific cultural and/or clinical bias that act as obstacles to pharmacogenomic implementation and proposes viable courses of action that might bring about change in cultural/medical habits.

## 1. Introduction

Individual genetic characteristics are known to influence drug response. The reasons may vary and include different adsorption capacities, absence of specific drug metabolic enzymes, or even polymorphisms regarding drug metabolism [1,2]. Additionally, therapeutic doses concentration in the blood are influenced by genetic matrix, causing extremely high or extremely low scenarios, and thus leading to serious side effects, including death [3]. Proper understanding requires the gathering of evidence and knowledge of a range of concepts and disciplines. 

Single gene analysis is performed by pharmacogenetic studies, to assess the influence of genetic factors. Usually, pharmacogenetic tests are used to detect the presence or absence of mutation in a given gene or chromosome. Despite the reliability of such tests, it has been well established that the pharmacological response of a particular drug may be due not only to a single gene but to a large number of genes [4]. The study of a full set of genes with pharmacological relevance thus emerged as field of particular interest resulting in the establishment of pharmacogenomics. The basic principles underlying pharmacogenomics are outlined in Figure 1. Briefly, two main genomic alterations occur. Firstly, poor metabolizers for a certain drug will generally display a phenotype of increased drug concentration or time of circulation of the drug in the bloodstream, something that might affect toxicity levels and the appearance of adverse reactions. This phenotype is commonly due to loss of function on the part of genetic loci. These genetic loci are responsible for the metabolizing enzymes such as gene deletions, mutations, and other genetic alterations (polymorphisms) and potentially result in the total or partial loss function on the part of these enzymes. On the other hand, the phenotype of rapid (and ultra-rapid) metabolizers is potentially due to increased production of the metabolizing enzymes perhaps as a result of a single or multiple duplication of the gene encoding such enzymes. In this case, drug concentration will fall below the therapeutic window, resulting in the absence of the desired clinical effect.

Pharmacogenomics is recognized by the Food and Drug Administration (FDA) as a highly significant player in the health area of response identification and is of use in avoiding adverse effects and viewing the optimization of administered doses. The FDA also considers pertinent the variability description of clinical drug response, as well as dosage dependence of specific genotype, the mechanism of action and the polymorphic targets, in drug inserts. Already, 120 recognized drugs present this information, including drugs in therapeutic areas such as oncology, psychiatry, endocrinology, or cardiology [5]. 

Within oncology, breast cancer is the most common type of cancer in women and the second leading cancer responsible for death. In Portugal—an EU country—approximately 6974 new cases are detected per year. Of these, 1748 cases are mortal rendering breast cancer the fifth largest cause of death in those individuals that have been detected and diagnosed [6]. This percentage does not correspond to the survival rates at five years provided by the American Cancer Society (ACS): 99% for early stage breast cancer (ESBC), 84% for locally advanced breast cancer (LABC), and 24% for metastatic breast cancer (MBC) [7].

Surgery, radiation therapy, chemotherapy, hormonal therapy, or some combination of these constitute, among other treatments, are common breast cancer therapies. The decision factors in prognosis and choice of the therapy most suited to individual cases include age, tumor/nodal/metastasis (TNM) staging variables, estrogen and progesterone receptor status, HER-2 gene stage, and pre or post-menopausal and vascular lymphatic space invasion [8]. Studies have shown that a fifth of all breast cancers appear before the age of 50, and approximately two thirds are estrogen receptor positive. After the age of 50, this increases to 80% estrogen receptor positive [9].

Adjuvant therapy, which includes chemotherapy, radiotherapy, hormonal therapy, or other targeted therapy, is performed after the initial primary treatment, i.e., after post-surgical therapy, aiming to reduce the risk of recurrence [10]. Hormonal therapy is a field where pharmacogenomics has a key role to play.

Decision factors in opting for hormonal therapy include selective estrogen receptor modulators (SERM), aromatase inhibitors (AI) and ovarian ablation. Hormonal therapy is also desirable for adjuvant therapy in postmenopausal women with estrogen positive breast cancer, five years of aromatase inhibitors, or two years of tamoxifen followed by three years of an aromatase inhibitor [11].

A number of authors claim that adjuvant therapy with an AI improves outcomes, as compared with tamoxifen, in certain premenopausal women with breast cancer [12]. However, the most recent randomized clinical trials (RCTs) comparing the AI, anastrozole, with tamoxifen as a means of treating estrogen receptor-positive cancers showed that the AI was at least as effective as tamoxifen, with differences in side-effect profiles [13,14]. 

Tamoxifen is a pro-drug of the SERM class. In the postmenopausal case, however, aromatase inhibitors are the drugs usually selected for hormone dependent cancers. Because of the decreased estrogen concentration levels, tamoxifen’s effectiveness depends of several factors: adherence to treatment, the metabolizer genotype CYP2D6, and the use of concomitant medication that can inhibit the conversion of tamoxifen into metabolites [15,16].

Tamoxifen is a weak anti-estrogen, with agonist properties, extensively metabolized into 4-hydroxytamoxifen, 4-desmethyltamoxifen, and endoxifen, three potent anti-estrogens. Tamoxifen has been used as therapy and prevention of relapse in patients with estrogen receptor or progesterone receptor positive [17]. Figure 2 illustrates tamoxifen action. 

According to the results of ATLAS trial—a large international clinical trial—authors have established that, for a number of women with breast cancer, taking adjuvant tamoxifen for 10 years after primary treatment leads to a greater reduction in breast cancer recurrences and deaths than taking the drug for only 5 years [18]. Accordingly, the risk of recurrence due to tamoxifen is reduced in 11.8% of cases and mortality for five years is also reduced, though by 9.2% of cases [19]. 

CYP2D6 is one of the most extensively studied drug-metabolizing enzymes and pharmacogenes. CYP2D6 alleles confer normal, decreased, or no activity and cause a wide range of activity among individuals and between populations. For some alleles, similar frequencies have been determined among African Americans, East Asians, and Europeans [20,21,22]. Genetic variations on CYP2D6 are associated with reduced concentration of endoxifen [1] due to reduced CYP2D6 enzyme activity [23]. Since endoxifen have about 100-fold more affinity for estrogen receptor than tamoxifen, the effectiveness of tamoxifen is affected by the decrease in the concentration of endoxifen. Consequently, this has major impact on clinical outcomes [11]. The CYP2D6 gene is highly polymorphic with over 100 catalogued alleles, and clinical CYP2D6 testing is increasingly accessible and supported by practice guidelines [24]. 

Although a large number of investigations into tamoxifen pharmacogenomics have been performed, there are still inconsistencies with regard to the association results between efficacy of tamoxifen and genetic polymorphism of CYP2D6. Some authors have demonstrated that such polymorphisms on CYP2D6 gene are not to be associated with survival in breast cancer patients [16,25,26]. Nevertheless several other studies have reported a significant association between the CYP2D6 genotypes and clinical outcome of breast cancer patients receiving the tamoxifen therapy in the adjuvant setting [11,17,27]. The CYP2D6 gene is highly polymorphic. Alleles have been described in detail and compiled at the Pharmacogene Variation (PharmVar) Consortium (Table 1).

When no variant is detected and is assumed to have normal enzyme activity, CYP2D6*1 is assigned. Such CYP2D6 allele is responsible for the normal metabolizer phenotype. There are a number of population differences among the incidence of some the alleles of CYP2D6. For example, alleles *3, *4, *6, and *41 are more frequent in Caucasians, *10 more common in Asians, and *17 more common in Africans [22,28].

Pharmacogenomics investigated the metabolism of tamoxifen, and the first studies provide reasons for optimism with regard to the CYP2D6 phenotype pharmacogenomics test and suggest that this test could identify patients that are nonresponsive or different-responsive to usual tamoxifen administration [29,30,31,32]. Indeed, the FDA approved the AmpliChip^®^ CYP450 which analyzes 33 variants of the CYP2D6 gene (in several settings) [33,34,35]. These perspectives do not collide with a modelling study suggesting that pro-tamoxifen extensive phenotype patients could receive equal benefit either from tamoxifen or an aromatase inhibitor [36]. 

The present analysis takes a specific clinical context: Portuguese women with breast cancer (BC), and evaluates the possible impact of genotyping tests for CYP2D6 relative to tamoxifen. This is an interesting context, since these tests have never been conducted in Portuguese hospitals or clinics and would all but provide a control sample for comparison with the studies conducted so far elsewhere. These latter studies have typically analyzed hybrid contexts of tamoxifen administration. The feasible introduction of new clinical practice regarding pharmacogenomics—namely the CYP2D6 case—would permit the establishment of health policy models for other cultural and international contexts.

Thus in the present study we used a British survey and report [37], regarding the relationship between economy, healthcare, and clinical effectiveness of a pharmacogenomics test for BC. By doing so we intend to analyze what is happening in Portugal in what concerns to the use of pharmacogenomics in comparison with other countries. Namely, we aim to investigate the use of CYP2D6 genotyping prior to tamoxifen screening.

## 2. Materials and Methods

Pharmacogenetics tests for CYP2D6 have never been undertaken in Portuguese hospitals. This population is mainly Caucasian but, nevertheless, possesses multicultural and genetic diversity, mainly as a result of influence from African and East European countries. 

The data obtained in Portuguese clinical context (seven public hospitals, one private hospital and a private laboratory), as well as the data collected by Fleeman et al., is presented in Table 2. As a result, comparative analysis between the present survey and the data presented by UK researchers is achieved with relative ease [37]. The survey investigated the application of CYP2D6 tests, number of tests per year, type of tested alleles, economic costs of applying the tests, and additionally, the application of two pharmacogenomics tests, the TaqMan^®^ and AmpliChip^®^ CYP450. These two tests are able to correctly identify the alleles while AmpliChip^®^ CYP450 is specifically approved by the FDA [38].

The validated survey by Fleeman formed the basis of the questionnaires developed and used to assess and understand Portuguese clinical practice in the effective use of hormonal therapy, especially with tamoxifen [37]. These questionnaires were applied as personal interviews with the clinical and oncology department directors of the largest Portuguese hospitals in terms of oncological patient number, across the seven public hospitals (three from north region and four from the south), the private hospital (in the south) and a private genetics laboratory, which has this genetic test available. The present results can be compared with the above-mentioned survey and study.

## 3. Results

This study comprises two stages: (1) a transversal analysis compares International pharmacogenomics context versus Portuguese current pharmacogenomics context; and (2) an investigation regarding reasons and motivation in pharmacogenomics implementation as used in Portuguese Clinical practice. 

### 3.1. Comparative Analysis in the International Pharmogenomics Context

The analysis of the data present in Table 2 indicates the absolute absence of CYP2D6 testing for patients treated with tamoxifen in a Portuguese context, whether that be in public hospitals or in the private laboratory. This contrasts with data collected in the UK, NLD, and USA laboratories, where CYP2D6 genotyping is applied, with one exception, the LGC (Middlesex, UK).

However, the three countries can be clearly differentiated regarding the number of applied tests: in the UK, the number ranges from 0 to 24, in NDL is in the order of 300, and in the USA, the one surveyed clinic reports 1500 cases. These numbers correspond to a one-year period. The cost per test is fluctuant as well: from £30 to £500 (US$45.49 to US$758.09; €39.28 to €654.71). Nevertheless, it should be noted that the costliest implementation of CYP2D6 testing, the LAB21 (Cambridge, UK), included the use of TaqMan^®^, along with Luminex^®^ and sequencing, which could account for the difference in price. Interestingly, the only medical UK institution in the study that performed zero tests, LGC Middlesex, does not use TaqMan^®^ either but uses a fluorescent probe named HyBeacon^®^. The varying use of different adjoining tests with CYP2D6 testing is recurrent in the five non-Portuguese laboratories, something that confirms this varying use is a highly influential factor in the indicated cost per test. Variations in the use of pharmacogenomics tests, from the most to the least expensive, include: (a) CYP2D6 testings, TaqMan^®^ and Luminex^®,^ and sequencing kit (£500; US$758.09; €654.71); (b) CYP2D6 and TaqMan^®^ (£291.73; US$442.32; €382); (c) CYP2D6 and a kit from Luminex Molecular Diagnostics (£289.74; US$439.30; €379.39); (d) CYP2D6 and Amplification—Refractory Mutation System and Scorpions technology^®^ (£30; US$45.49; €39.28). AmpliChip^®^ CYP450 use is not general, unlike TaqMan^®^ (with or without other adjoining tests). The AmpliChip^®^ CYP450 analyses the patient genotypes for cytochrome P450 (CYP) genes. It classifies individuals into four phenotypes by testing twenty-seven alleles. The combination of AmpliChip^®^ CYP450 test, approved by the FDA, and the definition of CYP2D6 as a valid biomarker means that CYP2D6 is a candidate for use as a successful pharmacogenomics test in clinical practice [20]. Only one laboratory, the Erasmus University Medical Centre (Rotterdam, The Netherlands), acknowledged using this test in its practice within a CYP2D6 testing context. In fact, this laboratory uses TaqMan^®^ analysis as duplicate to confirm the eight most prevalent alleles and the gene duplication.

### 3.2. Portuguese Clinical Practice and Pharmacogenomics Implementation

The above-mentioned absence of CYP2D6 testing in Portuguese clinical practice prompted the interest of this article’s researchers, leading to conduct interviews in order to understand the reasons that underlie such a state of affairs. The data collected is presented in Table 3. 

Unanimously, the institutions agreed that: (1) pharmacogenomics is of essential nature when it comes to predict possible reactions to drugs, and especially in the tamoxifen scenario; (2) this particular study of CYP2D6 genotyping has never been used in the Portuguese tamoxifen scenario; (3) there is absence of genetic testing before prescribing tamoxifen; (4) the clinical practice criteria for prescription is the sum of two factors, on one hand, the analysis of estrogen and progesterone receptors and Ki67 antigen with, on the other hand, data collected in question 2 of the present survey (a survey that assesses the importance of several factors: hormone dependent cancer, pre/post-menopausal, TNM grade, or identification of drugs resistance due to genetic polymorphism); (5) the appliance of genetic testing is discouraged.

Regarding this last point, we would like to quote one of those directors: “We don’t apply the test because there is no international recommendation for its implementation and the results of several clinical trials are still contradictory and therefore is no consensus on its use. Furthermore, its use is discouraged”.

One other reason given the directors is of an economic nature, as these tests are extremely expensive when compared with tamoxifen. This should be set in context of the known effectiveness of tamoxifen, which has been on the market since the early 70’s. This feedback suggests that time validated clinical practices—such as the usual administration of tamoxifen—minimize the need to introduce new medical habits, such as the pharmacogenomic pre-therapy testing, especially if such new medical habits are costly. In fact, the public in general is not informed of these possibilities, nor are the clinical professionals obliged to alert them to its presence.

However, the genetic laboratory offered another perspective. They considered the test both validated and useful its use in clinical practice remains complicated. In their opinion, a change of mentalities would be required were the product to be used and it remains the case that it is imperative that the cost-effectiveness ratio and clinical impact are assessed before introducing changes to the existing paradigm. They even stressed that, at present, although they offer the test to patients undergoing the CYP2D6 genotyping test in their laboratory, it could only be used by patients if the patient’s doctor specifically prescribed it This prescription could only be subject to common use after a change in medical paradigms.

Finally, the collected data regarding the alleles tested in the non-Portuguese laboratories indicated, inter alia, absence of testing and different laboratory choice of tested alleles. This particular aspect will be further explored in the next section (Section 4: Discussion).

## 4. Discussion

Fleeman et al. intended to clarify the current clinical practice regarding CYP2D6 testing for patients treated with tamoxifen, choosing to apply a questionnaire to laboratories recognized as offering these specific testing options [37]. Additionally, the laboratories present in the UK survey are involved in research activities and, as a result, they have state-of-the-art techniques available. It is also of no minor interest that these laboratories, covering three different countries: UK (three laboratories), The Netherlands (one laboratory), and USA (one laboratory) are of a trans-national nature. Comparison was more effective due to wide variety of national contexts. One of the major limitations of this survey is that it is somewhat old (published in 2011). However, to our knowledge, there has been no other similar survey, nor do we have evidence that any new survey has been undertaken since 2011. This being said, the main strength of survey remains its international nature involving several European and US countries.

As stated in the previous section, the data presented in Table 1 and Table 2 yielded the null presence of CYP2D6 genotyping tests for tamoxifen, in public sector, and therefore, the complete absence of systematic public genetic control of BC patients regarding their response to tamoxifen. In fact, this kind of test is available only in Portuguese private sector, while the LGG, Middlesex, UK is the only international laboratory that does not offer such testing.

The variability either in number of tests performed and the cost of the tests themselves also indicates factors beyond simple clinical reasons in the policies adopted. 

Such heterogeneous use of tests promoting CYP2D6 genotyping in the sample analyzed of laboratories (except the Portuguese scenario) also indicates that economic factors are key in clinical decision-making. An absence of international recommendations regarding these genetic tests further contributes to the high cost introducing biomedical technology. Indeed, only the Erasmus Medical Centre (Rotterdam, The Netherlands) recognized the use of these tests in systematic clinical practice. Similarly, this Centre uses the TaqMan^®^ as a second test to confirm the eight most prevalent alleles and possible duplication of genes.

Deeper analysis of the possible importance of alleles tested in non-Portuguese laboratories is to be found in the data presented in Table 4. 

From analysis of the Table 4, it is clear there is a somewhat wide range of various alleles testing, something that is justified, in line with the data presented in Table 2, by the heterogeneity of research studies or clinical trials [11,16,17,39]. Some of the most frequently found alleles are CYP2D6*4, *10, and *41, a finding that is in accordance with previously multi-ethnic population genomics studies. These found that CYP2D6*4, *10, *17, *29, and *41 are among the most frequent and also have similar frequencies among African Americans, East Asians, and Europeans subjects [20]. Nevertheless, other studies mention that alleles *4 is more frequent in Caucasians, *10 is more frequent in Asians, and that *17 is more commonly found in Africans [28].

Such a heterogeneous array of studies and trials explains the perceived lack of consensus regarding the CYP2D6 testing and subsequent inclusion criterion, doses, the duration of therapy, and the prospect of additional therapeutics in the form of chemotherapy and/or radiotherapy. 

From the data presented in Table 4, the variability in sheer number of tested alleles versus time of research study/clinical trial implementation stands out. The initial research study/clinical trial was implemented in the first decade of present century, included five tested alleles [39]. In the second decade, the number of tested alleles dropped from nine [17], to eight [16], and the latest research study/clinical trial, returned to the initial number of five, although with different tested alleles [10]. However, there are recurrent tested alleles in all research studies and clinical trials, namely *4, *10 and *41. These specific alleles are associated with loss of function (characteristic of poor metabolizers) or decreased activity (related with the intermediate metabolizer phenotype) [39]. 

Taking into account the fact that TaqMan^®^, for example, requires individual testing of each allele, it is clear to what extent cost and testing time-span with regard to the number of tested alleles remains significant. This may explain the reduction in the number of tested alleles through time, as more research studies and clinical trials have been implemented.

It should be emphasized that it is through the analysis of the different alleles that the individual phenotyping of the patient is carried out. There is considerable inter-ethnic variability in the incidence of different alleles with the result that the probability of a patient being poor metabolizer—should the test be positive (positive predictive values)—varies with patient’s ethnicity. It follows that there is a likely probability of genotyping not identifying the correctly alleles associated with poor metabolization of tamoxifen (false negatives) in certain populations and this might represent a significant limitation when it comes to AmpliChip^®^ application. 

## 5. Conclusions

This research study proposed itself to investigate the impact of genetic testing for CYP2D6 in relation to tamoxifen in the management of women with breast cancer, within a specific hospital European Union (EU) country context. The country, Portugal presents an interesting case for comparison in international pharmacogenomics context. This is especially the case of CYP2D6 testing prior to tamoxifen therapeutics. This is because there are no implemented measures based on pharmacogenomics analysis prior to therapy. Changing clinical paradigms involves assessment of several factors and a country with a clinical context as that of Portugal might function as a sample control in such analysis.

Regarding Portuguese clinical perception as is generally understood by clinical practitioners, Portugal is no different from most countries in Europe and America when it comes to considering the added value of these tests. From the analysis of Portuguese clinical situation, this research proposes a prospective methodology for a change in clinical paradigm, regarding the practice of including these tests. The following factors were identified and must be addressed (1) although there is no evidence and no ongoing studies that could demonstrate clinical utility of pharmacogenomic testing—primarily due to the lack of clinical validity—analytical validation should be developed, in order to achieve more homogeneous array of results (failing which there will be suspicion concerning the value of large scale implementation of the tests); (2) this analytical validation must be followed by a clinical validation; (3) the economic impact of the introduction of these genetic tests must be assessed and deemed acceptable when compared with the present therapies used. In this particular assessment, two further factors should be taken in account: (a) cost-effectiveness ratio of test implementation; and (b) adjoining alleles pre-testing (which tests and which alleles); (4) the direct costs and test benefits per se are not enough to correctly establish a clear idea of economic impact. 

With regard to future perspectives, we believe it would be interesting to survey the laboratories once again after 10 years (2021) and to observe relevant changes that have taken place over this period. However, we believe that are more important than issues to be addressed. One of the main disadvantages of using commercial chips is the limited number of allele variants which can be detected. With the major development of next-generation sequencing (NGS) platforms that are faster and affordable than ever before, it is possible to develop hypothetical predictions about the impact of this whole-genome based approaches in the daily pharmacogenomics routine. Another key question that needs to be answered before clinical validation and that is whether the cost-effectiveness of genetic testing for CYP2D6 as a management option in women with breast cancer is viable. One must not overlook the possible inclusion in regular clinical practice of genomic tests, such as the twenty-one gene recurrence score. These predictive tests are used to evaluate adjuvant chemotherapy in breast cancer. These tests may also optimize the management of chosen adjuvant therapies. Future analysis of this scenario in Portuguese hospital context might well form the basis to interesting further studies.

Until scientific evidence improves unequivocal clinical decision-making guidelines, the clinical use of this and other genetic tests should be taken with caution. This is the case not only with regard to interpretation, but also in terms of application, in so far as the management of health care services is concerned. 

## Figures and Tables

**Figure 1 medicina-55-00344-f001:**
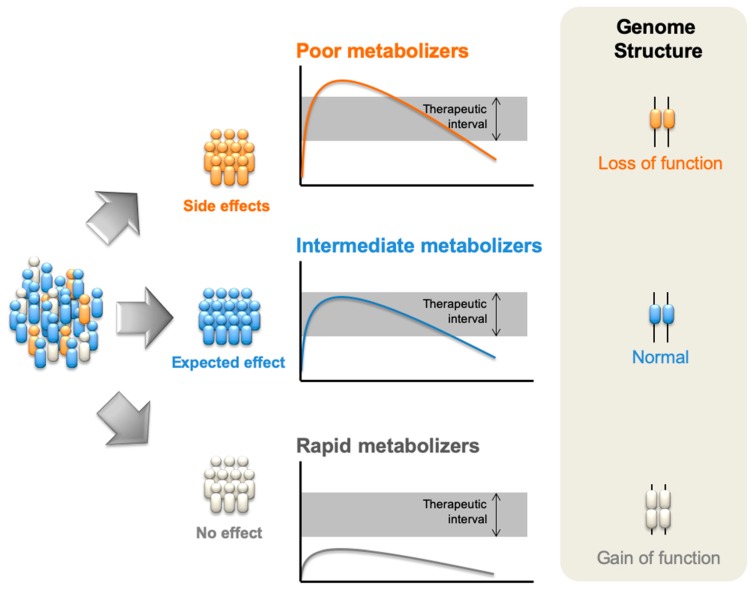
Principles of pharmacogenomics. Drugs may be metabolized as expected and the active drug will be present in the organism at an ideal level of concentration and/or during the expected amount of time (center); However, in some genomic alterations such as mutations, deletions, or polymorphisms with loss of function in the biochemical systems that metabolize the drug, there is a decrease in enzyme activity. This mean that drugs will be presented at higher concentration doses and/or during a greater length of time (top). Alternatively, genomic duplications and gain of function polymorphisms will increase the enzymatic activity, leading to lower active drug concentration (bottom).

**Figure 2 medicina-55-00344-f002:**
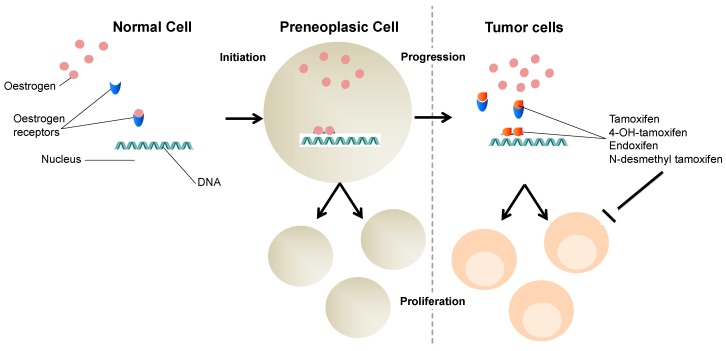
Tamoxifen action. Tamoxifen acts as an estrogen competitor. Estrogens are lipid hormones that are able to pass through membranes and bind to their receptors in cytoplasm. The complex estrogen–estrogen receptor enters the nucleus and dimerizes with another similar complex. Estrogens have a transcription mode of action contributing, among others, to cell proliferation. Breast cancer cells have an increased expression of estrogen receptors and tamoxifen, and its potent metabolites, such as 4OH-tamoxifen and endoxifen. Those will bind to them, in order to, hopefully, inhibiting cancer cells proliferation.

**Table 1 medicina-55-00344-t001:** Activity Status of Selected CYP2D6 alleles.

Allele Type	CYP2D6 Allele
Normal function	*1, *2, *7, *8, *33, *35
Decreased function	*9, *10, *14A, *14B, *17, *29, *41
No function	*3, *4, *5, *6, *7, *8, *11, *12, *13, *15, *19, *20, *21, *36, *38, *40, *42

Adapted from Dean, L. (2019) [28].

**Table 2 medicina-55-00344-t002:** Data presented in The Clinical Effectiveness and Cost-Effectiveness of Genotyping for CYP2D6 for the Management of Women with Breast Cancer Treated with Tamoxifen: a systematic review (Fleeman et al., 2011) and from Portuguese implemented survey.

Question	Laboratory						
	LAB21 (Cambridge, UK)	Mayo Clinic (Rochester, MN, USA)	DxS (Manchester, UK)	LGC (Middlesex, UK)	Erasmus University Medical Centre (Rotterdam, The Netherlands)	Inquired Hospitals in PT (*n* = 7)	Private Laboratory PT
How many requests per year do you get for CYP2D6 testing for TAM?	Overall number is small but increasing, last 12 months: 12 requests.	1500 tests per year	Two per month	No tests	300 tests per year	No tests	Unavailable data
When you do clinical testing for CYP2D6 which alleles do you test?	*2, *2A, *3, *4, *6, *7, *8, *9, *10, *11, *12, *17 and *N*	*2 through *12, *14, *15, *17, *41	NS	No tests	*3, *4, *5, *6, *9, *10, *41	No tests	Unavailable data
Do you use TaqMan^®^?	Yes, along with a kit from Luminex^®^ and sequencing	No, use a kit from LuminexMolecular Diagnostics	No, use Amplification—Refractory Mutation System and Scorpions technology^®^ (DxS Surrey, UK)	No, use a fluorescent probe called HyBeacon^®^ (LGC Middlesex, UK)	Yes	No	Unavailable data
Do you offer AmpliChip^®^ CYP450 testing?	No	No, it is too costly	No	No	Yes, and TaqMan^®^ analysis as duplicate to confirm the eight most prevalent alleles and the gene duplication	No	Unavailable data
How much do you charge for a CYP2D6 test? *	£500 US$758.09 €654.71	£289.74 US$439.30 €379.39	£30 US$45.49 €39.28	NS	£291.73 US$442.32 €382	___	£107.24 US$164.05 €145

* Monetary conversion using trading currency from the Banco de Portugal (Bank of Portugal) in 20 January 2015.

**Table 3 medicina-55-00344-t003:** Data collected through interviews of Portuguese public and private clinical and oncology service directors.

Parameters/Questions	Answers
Role of Pharmacogenomics	Essential in predicting possible reactions to drug
General Genetic Testing	Not used in Portuguese Tamoxifen scenario
Prescription Criteria	Sum of two factors: (1) the analysis of estrogen and progesterone receptors and Ki67 antigen; and (2) data collected in the patient’s clinical history (for example, hormonal dependent cancer)
Application of Genetic Testing	Discouraged, since: (1) there is no international recommendation for its implementation; and (2) the results of several clinical trials are still contradictory.
Public Awareness	Genetic testing more expensive than general prescription of tamoxifen
Private/Public Perspectives in Genetic Testing	Genetic laboratory considered it both validated and useful; public hospitals considered tamoxifen without pre-testing a validated practice.

**Table 4 medicina-55-00344-t004:** Tested alleles before CYP2D6 testing per research studies.

Study	Alleles Tested
Schroth et al., 2007	*3, *4, *5, *10, *41
Kiyotani et al., 2010	*4, *5, *6, *10, *14, *18, *21, *36, *41
Regan et al., 2012	*2, *3, *4, *6, *7, *10, *17, *41
Goetz et al., 2013	*3, *4, *6, *10, *41
Hertz et al., 2017	*2, *3, *4, *6, *10, *41
Sanchez-Spitman et al., 2019	*3, *4, *5, *6, *7, *8, *9, *10, *11, *12, *13, *14A, *15, *17, *19, *20, *29, *36, *40, *41

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
