# Peer review of "Pharmacogenomics, CYP2D6, and Tamoxifen: A Survey of the Reasons Sustaining European Clinical Practice Paradigms"

_medicina, 2019, doi:10.3390/medicina55070344_

Round 1

Reviewer 1 Report

Major:

The original survey was done in 2011, prior to the public presentation of the analyses of BIG 1-98 (Pubmed ID: 22395644) and ATAC (PMID: 22395643), neither of which confirmed the association between CYP2D6 genotype and tamoxifen treatment outcomes. Since then, multiple prospective trials have failed to establish an association between endoxifen concentration or CYP2D6 genotype and tamoxifen treatment outcomes (PMIDs: 30676859, 29459457). While there may have been some preliminary evidence supporting CYP2D6 testing to guide tamoxifen treatment in 2011, there is far less evidence and interest to support doing so now. Therefore, comparison of these survey findings in a new lab to original survey findings from other labs in 2011 is scientifically invalid. It may be interesting to go back and re-survey the same institutions that were surveyed in 2011 to see how their perspective and practice has changed, but that’s a completely different study.

Minor:

Figure 1: This is a great figure, however, the pharmacokinetic curves depict differences in absorption/distribution whereas the classical pharmacogenetic effect is on metabolism (the slope of the elimination phase). It would be better to change these lines so they depict differences in elimination. These curves also suggest that Cmax is the critical variable for determining treatment response, but this is very seldom the case. If you could depict steady state and show slow metabolizers above therapeutic interval and fast metabolizers below, that would be better. Finally, there are consensus terms that are used including “Poor Metabolizers” and “Rapid Metabolizers” or “Normal Metabolizers.” We don’t use the terms “fast” or “slow” to describe metabolic phenotypes.

Line 66: The incidence of breast cancer (4500) is not directly related to the number of deaths per year (1500) as it is not the case that 1500 of the 4500 patients diagnosed that year will die within that 12 month period (hence 5-year survival estimates).

Line 86: The recent SOFT and TEXT trials showed that aromatase inhibitors are superior to tamoxifen with ovarian suppression in pre-menopausal patients (See PubMed ID: 24881463)

Line 107: The hypothesis about CYP2D6 genetics and activity well predates this reference.

Line 156: The statement that labs involved in research do not have access to state-of-the-art techniques is mistaken. Research is, essentially by definition, ahead of clinical practice when it comes to techniques.

Pg 7 line 3: These numbers represent a single lab, not the whole country.

Pg 7 line 28: I would be amazed, and honestly disappointed, if the institutions unanimously endorsed that pharmacogenomics is essential for tamoxifen. This is NOT verified (see comment above). That’s likely why the “appliance of genetic testing is discouraged” (line 36 and quote 37-40).

Line 110: Analytical validation refers to the ability of the genetic test to produce an accurate genetic result. For single SNP (Taqman) or multi-SNP (Amplichip) tests this is not an issue at all. The CYP2D6-tamoxifen association has not documented “Clinical validity” meaning there is no confirmation that genotype predicts treatment outcomes (as described above). More troublingly for the rationale for this survey, it requires evidence of “Clinical Utility” to warrant pre-emptive genotyping to guide treatment. There is no evidence and no ongoing studies that could demonstrate clinical utility, primarily due to the lack of clinical validity.

Author Response

Major:

The original survey was done in 2011, prior to the public presentation of the analyses of BIG 1-98 (Pubmed ID: 22395644) and ATAC (PMID: 22395643), neither of which confirmed the association between CYP2D6 genotype and tamoxifen treatment outcomes. Since then, multiple prospective trials have failed to establish an association between endoxifen concentration or CYP2D6 genotype and tamoxifen treatment outcomes (PMIDs: 30676859, 29459457). While there may have been some preliminary evidence supporting CYP2D6 testing to guide tamoxifen treatment in 2011, there is far less evidence and interest to support doing so now. Therefore, comparison of these survey findings in a new lab to original survey findings from other labs in 2011 is scientifically invalid. It may be interesting to go back and re-survey the same institutions that were surveyed in 2011 to see how their perspective and practice has changed, but that’s a completely different study.

AUTHORS: First of all, we are most honored to have the opportunity to answer to reviewer 1 comments. For the general comments, this reviewer as brought an important insight on the manuscript, which will, certainly improve the quality of the present work.

The original papers that reviewer one mentioned are of major importance. Neverthless, and without undervalue, Goetz, 2013 (PMID:23213055), Kiyotani, 2010 (PMID: 20124171), and Ramón y Cajal, 2010 (PMID: 19189210) argue otherwise. We agree that this topic is still a 'hot topic' in the pharmacogenomics of breast cancer, and thus, confirming the importance of the present work.

Now we will answer to all the questions that reviewer gently pointed-out. In the main text, all the text highlighted in green are alterations to address questions made by reviewer one. 

Minor:

Figure 1: This is a great figure, however, the pharmacokinetic curves depict differences in absorption/distribution whereas the classical pharmacogenetic effect is on metabolism (the slope of the elimination phase). It would be better to change these lines so they depict differences in elimination. These curves also suggest that Cmax is the critical variable for determining treatment response, but this is very seldom the case. If you could depict steady state and show slow metabolizers above therapeutic interval and fast metabolizers below, that would be better. Finally, there are consensus terms that are used including “Poor Metabolizers” and “Rapid Metabolizers” or “Normal Metabolizers.” We don’t use the terms “fast” or “slow” to describe metabolic phenotypes.

 AUTHORS: We agree and have changed it.

Line 66: The incidence of breast cancer (4500) is not directly related to the number of deaths per year (1500) as it is not the case that 1500 of the 4500 patients diagnosed that year will die within that 12 month period (hence 5-year survival estimates).

 AUTHORS: We agree and have reviewed it.

Line 86: The recent SOFT and TEXT trials showed that aromatase inhibitors are superior to tamoxifen with ovarian suppression in pre-menopausal patients (See PubMed ID: 24881463)

 AUTHORS: Thank you for your insight. We have discussed that subject and improved it.

Line 107: The hypothesis about CYP2D6 genetics and activity well predates this reference.

 AUTHORS: We agree and have reviewed it and clarified it.

Line 156: The statement that labs involved in research do not have access to state-of-the-art techniques is mistaken. Research is, essentially by definition, ahead of clinical practice when it comes to techniques.

AUTHORS: We agree and have corrected. 

Pg 7 line 3: These numbers represent a single lab, not the whole country.

 AUTHORS: We agree and have it is already mentioned.

Pg 7 line 28: I would be amazed, and honestly disappointed, if the institutions unanimously endorsed that pharmacogenomics is essential for tamoxifen. This is NOT verified (see comment above). That’s likely why the “appliance of genetic testing is discouraged” (line 36 and quote 37-40).

AUTHORS: We understand that point-of-view since it still a controversial topic. In fact, that topic has reviewed and extended, so we believe that it supports your vision in this topic (lines 105-110).

Line 110: Analytical validation refers to the ability of the genetic test to produce an accurate genetic result. For single SNP (Taqman) or multi-SNP (Amplichip) tests this is not an issue at all. The CYP2D6-tamoxifen association has not documented “Clinical validity” meaning there is no confirmation that genotype predicts treatment outcomes (as described above). More troublingly for the rationale for this survey, it requires evidence of “Clinical Utility” to warrant pre-emptive genotyping to guide treatment. There is no evidence and no ongoing studies that could demonstrate clinical utility, primarily due to the lack of clinical validity.

AUTHORS: We agree and emend it.

Reviewer 2 Report

This is an interesting work trying to investigate he impact of genetic testing for CYP2D6 in relation to tamoxifen in the management of women with breast cancer, in Portugal, compared with other countries. The manuscript is well English written, However, I have the following observations:

Major comments

The introductory section of the manuscript describes the background information about tamoxifen pharmacogenomics and includes a figure (figure 1) that is a very recurrent scheme, is not new, and there is not reference to a previous figure. Besides, it is not complete or adequate, because assigns deletions and SNPs to slow metabolizers, which is not always so. It is known that some SNPs can produce rapid metabolizers. Moreover the description of deletion should be with a discontinuous line. In spite of my observations, I believe this figure should be deleted.

There is a lacking aspect in the introduction. It is necessary to include a discussion about ethnic variations in the frequencies of CYP2D6 alleles, considering that authors compare CYP2D6 test with other countries of the region.

In the description of tamoxifen metabolism (lines 91-98), the authors do not include N-desmethyl tamoxifen, which is one of the active metabolites of tamoxifen (see PharmGKB web page).

The Materials and Methods section is incomplete, there are a number of lacking specifications, i.e, characteristics of the questionnaire/survey, institutions and professional participants, etc.

Some of the paragraphs of the results belong to the methods section (e.g lines 149-152, lines 160-163) and some others to the discussion (e.g. lines 153-159). The authors must reorganize it.

References are very old, the newest one is from 2013!.

Minor Comments

Figure 2 should be completed, it is very basic, and proliferation stage is not indicated.

In lines 112-119, the authors should discuss about sensibility and specificity of the AmpliChip®P45O. It is well known that both parameters, and also the predictive value, change when the test is applied to different populations.

Author Response

Major comments

The introductory section of the manuscript describes the background information about tamoxifen pharmacogenomics and includes a figure (figure 1) that is a very recurrent scheme, is not new, and there is not reference to a previous figure. Besides, it is not complete or adequate, because assigns deletions and SNPs to slow metabolizers, which is not always so. It is known that some SNPs can produce rapid metabolizers. Moreover the description of deletion should be with a discontinuous line. In spite of my observations, I believe this figure should be deleted.

AUTHORS: First of all, we are would like to thank you for the opportunity to answer to these comments. Reviewer two has underlined some major features that will be an important asset for this manuscript.

All the changes that we have made in the manuscript which were a direct answer to reviewer two, we have highlighted in cyan.

Regarding figure 1, reviewer one is in agreement with the authors about the relevance of that image. Not despising your comment, we improved such image in order to, hopefully, become of your accordance.

There is a lacking aspect in the introduction. It is necessary to include a discussion about ethnic variations in the frequencies of CYP2D6 alleles, considering that authors compare CYP2D6 test with other countries of the region.

AUTHORS: Thank you for this important input. We not only have altered in the introduction but also went further and have mentioned the importante of ethnicity in the discussion.

In the description of tamoxifen metabolism (lines 91-98), the authors do not include N-desmethyl tamoxifen, which is one of the active metabolites of tamoxifen (see PharmGKB web page).

AUTHORS: We agree and changed it, both in text and in figure 2.

The Materials and Methods section is incomplete, there are a number of lacking specifications, i.e, characteristics of the questionnaire/survey, institutions and professional participants, etc.

AUTHORS: Regarding this topic: We mentioned that we used the same questionnaire published by Fleeman et al. (2011). By the sake of anonymity, we must not mention the name of both institutions or professionals. However, we mentioned that all the respondents were directors of department (Oncology). 

Some of the paragraphs of the results belong to the methods section (e.g lines 149-152, lines 160-163) and some others to the discussion (e.g. lines 153-159). The authors must reorganize it.

AUTHORS: We agree and changed it.

References are very old, the newest one is from 2013!.

AUTHORS: We agree and improved.

Minor Comments

Figure 2 should be completed, it is very basic, and proliferation stage is not indicated.

AUTHORS: We thank you for your comment. It is intended to be that simple.

In lines 112-119, the authors should discuss about sensibility and specificity of the AmpliChip®P45O. It is well known that both parameters, and also the predictive value, change when the test is applied to different population

AUTHORS: Thank you for that question. But the diagnostic tests parameters indeed vary deeply in different populations. Also, there are little and sparsely information among literature. 

Reviewer 3 Report

There were several grammatical errors which at times made the manuscript difficult to follow. I have included a few of these errors listed below. Of note: 

Line 39: I am confused by the use of the "rehabilitee" as it does not fit with the context.

Line 55: I am unsure what is meant by "This state organism" 

Line 67: I believe you meant to say "tests" here instead of "testes"

Author Response

AUTHORS: First of all, thank you so much for your comments.

The alterations reagarding your comments are highlighted in pink on the manuscript. 

Line 39: I am confused by the use of the "rehabilitee" as it does not fit with the context.

AUTHORS: We agree... it was a mistake. We intended to say: reliability.

Line 55: I am unsure what is meant by "This state organism" 

AUTHORS: We agree and corrected.

Line 67: I believe you meant to say "tests" here instead of "testes"

AUTHORS: We also have changed this.

Reviewer 4 Report

Overall review: The article overall discusses an important pharmacological prodrug (tamoxifen) as it is used to treat breast cancer in CYP2D6-amenable patients. The article can be strengthened, however, with additional and more recent evidence on how tamoxifen influences or drives clinical outcomes (e.g. survival rates). In general, the article could benefit from citing more recent and relevant work. The authors make claims about studies that provide evidence on the pharmacogenomic role of CYP2D6, but do not cite enough studies to support some of their statements. On a minor note, the paper as it is written can be slightly improved in terms of grammar and use of English language (for example, with their reference to the FDA, I believe the term “organism” as it is used should be replaced with “organization”). Some specific comments (per line prior to table 1): The first sentence in the abstract can be revised for clarity or maybe even moved into the Background and objectives section in the abstract to make the purpose of the study clearer to the reader. Line 60-61: Controversy around the use of vaccines is being compared to a seemingly reluctant or slow uptake of pharmacogenomics in standard clinic practice. I am not sure if this is a fair comparison, as vaccines became controversial due to poor interpretation of scientific evidence, which was exacerbated by the media; this is contrary to pharmacogenomics being a nascent field with also nascent and evolving causational evidence. Lines 64-68: It is mentioned that, in Portugal, 4500 new breast cancer cases are detected per year, with 1500 of those cases being mortal. The five-year survival rates are then reported, according to the American Cancer Society for early stage breast cancer, locally advanced breast cancer, and metastatic breast cancer. I was left wondering what these five-year survival rates are in Portugal for each of these categories. I encourage the authors to include this information if they have it or can retrieve it. Also, the authors should cite more and recent studies, some of which are publicly accessible on PubMed Central, that explain survival rates in breast cancer patients. For example, the following population study published in 2017 argues with evidence from a univariate analyses that the CYP2D6 genotype (specifically, CYP2D6 polymorphisms *2, *3, *4, *6, *10, *41 and copy number variants) is not associated with recurrence free survival in breast cancer patients (p>0.2): Hertz DL, Kidwell KM, Hilsenbeck SG, et al. CYP2D6 genotype is not associated with survival in breast cancer patients treated with tamoxifen: results from a population-based study. Breast Cancer Res Treat. 2017;166(1):277–287. doi:10.1007/s10549-017-4400-8 Another study supporting this conclusion: Ahern TP, Hertz DL, Damkier P, et al. Cytochrome P-450 2D6 (CYP2D6) Genotype and Breast Cancer Recurrence in Tamoxifen-Treated Patients: Evaluating the Importance of Loss of Heterozygosity. Am J Epidemiol. 2017;185(2):75–85. doi:10.1093/aje/kww178 Line 93: This should be written as “oestrogen receptor or progesterone receptor positive” versus as it is currently written. Line 96-97: The authors mention that “according to several clinical trials, the risk of recurrence due to tamoxifen is reduced…” but only one study is referenced that is a qualitative study (versus a perhaps randomized clinical trial). Please add additional references to relevant clinical trials with significant findings. Line 112-113: The authors mention that the metabolism of tamoxifen has been investigated, but only reference one study from 2005. I encourage the authors to cite more studies, particularly studies used to justify the FDA’s approval of AmpliChip. Line 118-119: The authors explains that studies showing mixed results about tamoxifen have led to contradictory conclusions, but only reference one study. As with my comment for statements made in lines 112-113, the authors should cite more studies to substantiate the background of their work present. Lines 123-127: The authors explain that pharmacogenomic tests for CYP2D6 have never been implemented in Portuguese hospitals and that, because of this, their Portuguese study population of women with breast cancer can serve as a control sample (which I think is the more novel aspect of this study). It would add value to also explain the possible racial/ethnic origins of Portuguese breast cancer patients, as genotype could certainly depend on this. For example, Yang et al. mention that “majority of Sephardic Jews (SJ) currently reside in Spain and Portugal, with smaller populations in Israel, the USA, and other countries” and that “common multi-ethnic variants in key drug metabolism genes (e.g., ABCB1, CYP2C8, CYP2C9, CYP2C19, CYP2D6, NAT2) have also been detected in the AJ and other Jewish groups.” Citation is here- Yang Y, Peter I, Scott SA. Pharmacogenetics in Jewish populations. Drug Metabol Drug Interact. 2014;29(4):221–233. doi:10.1515/dmdi-2013-0069 Also see- Naranjo, M.E.G., Rodrigues-Soares, F., Penas-Lledo, E.M., Tarazona-Santos, E., Farinas, H., Rodeiro, I., Teran, E., Grazina, M., Moya, G.E., Lopez-Lopez, M. and Sarmiento, A.P., 2018. Interethnic Variability in CYP2D6, CYP2C9, and CYP2C19 Genes and Predicted Drug Metabolism Phenotypes Among 6060 Ibero-and Native Americans: RIBEF-CEIBA Consortium Report on Population Pharmacogenomics. Omics: a journal of integrative biology, 22(9), pp.575-588. Line 149-150: The authors initially state in lines 138-141 that interviews were conducted with directors of seven public hospitals, one private hospital, and a private genetics laboratory (n=9). However, here they state that their data was obtained from seven public hospitals and a private laboratory (n=8). Please clarify. More comments (per line after table 1): Results: It is interesting that the USA received 1500 requests per year for CYP2D6 testing for TAM, a much larger number than the other countries. Was a reason for this observation explored in this study? Table 3 presents four studies published between 2007-2013 that have tested CYP2D6 polymorphisms or “alleles” and use this table to demonstrate a heterogeneity of research studies in this area. I think, however, that it would be wise to include additional (2-3) and more recent studies in this table to sufficiently show heterogeneity/variability and to also see any recent or present-day trends in the number of alleles investigated since 2013. For example, RIBEF-CEIBA Network Consortium was established among researchers in Ibero-Spain, Portugal, and Latin America to study population pharmacogenomics of their populations. Have this group or other researchers published any relevant and recent work?

Author Response

Thank you so much for all your comments that will greatly improve our work.

We have addressed all your commentaries and suggestions.

Since there are several reviewers, we have highlighted the alterations to all your comments in yellow.

Only three aspects should be clarified.

1) Regarding this statement:

"The authors explain that pharmacogenomic tests for CYP2D6 have never been implemented in Portuguese hospitals and that, because of this, their Portuguese study population of women with breast cancer can serve as a control sample (which I think is the more novel aspect of this study). It would add value to also explain the possible racial/ethnic origins of Portuguese breast cancer patients, as genotype could certainly depend on this. For example, Yang et al. mention that “majority of Sephardic Jews (SJ) currently reside in Spain and Portugal, with smaller populations in Israel, the USA, and other countries” and that “common multi-ethnic variants in key drug metabolism genes (e.g., ABCB1, CYP2C8, CYP2C9, CYP2C19, CYP2D6, NAT2) have also been detected in the AJ and other Jewish groups.” Citation is here- Yang Y, Peter I, Scott SA. Pharmacogenetics in Jewish populations. Drug Metabol Drug Interact. 2014;29(4):221–233. doi:10.1515/dmdi-2013-0069 Also see- Naranjo, M.E.G., Rodrigues-Soares, F., Penas-Lledo, E.M., Tarazona-Santos, E., Farinas, H., Rodeiro, I., Teran, E., Grazina, M., Moya, G.E., Lopez-Lopez, M. and Sarmiento, A.P., 2018. Interethnic Variability in CYP2D6, CYP2C9, and CYP2C19 Genes and Predicted Drug Metabolism Phenotypes Among 6060 Ibero-and Native Americans: RIBEF-CEIBA Consortium Report on Population Pharmacogenomics. Omics: a journal of integrative biology, 22(9), pp.575-588. "

A study from Frontiers Genetics "Portuguese crypto-Jews: the genetic heritage of a complex history" has referred that the actual Jewish population in Portugal accounts for 546 people. Since Portugal has more than 10.000.000 people we believe that it would be an interesting academic fact, but would not contribute for the scope of the work. If you believe that it should be mentioned we happily introduce this fact on the manuscript. However, we improved materials and methods mentioning that Portuguese population is a caucasian population with other ethnics groups mainly from  African and East European countries.

2) Regarding the statement:

"...1500 requests per year for CYP2D6 testing for TAM, a much larger number than the other countries. Was a reason for this observation explored in this study?

The question is also very interesting but we believe that dropped out the scope of the study.

3) And finally, regarding the "RIBEF-CEIBA Network Consortium was established among researchers in Ibero-Spain, Portugal, and Latin America to study population pharmacogenomics of their populations. Have this group or other researchers published any relevant and recent work?

As far as we know there are no other studies regarding this topic. 

Round 2

Reviewer 1 Report

Thank you for resolving my minor reviewer comments. Unfortunately, I don't see a meaningful response to my major criticism: the original survey was conducted at a time (2011) when there was much stronger rationale for considering CYP2D6 guided tamoxifen. Since then, the intellectual basis for this has diminished, so there is no scientific value in comparing study results from 2011 to study results now in a different lab and country. I again would recommend you consider re-surveying the labs in the original survey and comparing their changes in practice over time if you want to make a meaningful contribution to the literature. 

Author Response

We kindly wish to thank you for your comments and constructive criticism. 

When we applied the survey in Portugal, it was a task that took about 3 months only scheduling the interviews with the directors all over the country. We cannot imagine how long should it take to manage to schedule and interview all of the internationals hospitals / labs mentioned in survey that was the starting point for the study in our country.

However, we accept the challenge of include that suggestion in our personal future goals.

Reviewer 2 Report

I appreciate the authors considered my observations regarding the text modifications and references, i feel the manuscript has been considerably improved. However, In spite of these modifications i still feel the manuscript need changes:

The abstract and Introductory section have been changed, but i think the abstract still needs to be improved, first line about CYP2D6 is extremely poor.

The figure 1 have been improved in order to overcome the reviewer’s comments. I proposed to delete it, but the authors want to maintain it in the text, however there is not reference for it, considering is not new. Moreover the description of gene deletion has not been incorporated, as requested. Please consider it.

Please indicate the proliferation stage (as text) in figure 2.

In relation to my comment that the authors should discuss about sensibility and specificity of the AmpliChip®P45O, because It is well known that both parameters, and also the predictive value, change when the test is applied to different populations the authors indicate that the diagnostic tests parameters vary deeply in different populations and there are little and sparsely information among literature. In this respect i think it is necessary  at least to hypothetize about it and its limitations (https://www.sciencedirect.com/topics/medicine-and-dentistry/amplichip), particularly in different ethnicities.

Author Response

We kindly wish to thank you for your comments. 

We hope that now we have fully addressed to all of them.

Our inputs in the main text highlighted in yellow refers to the improvements addressed to your comments.

Reviewer 4 Report

The manuscript has been significantly improved. However, some areas remain a bit unclear, such as the abstract. Also, the Fleeman et al reference and findings are a bit old (from 2011), which makes me question the validity of the comparison being made in the study. The authors should explicitely state and explain this limitation in their discussion and suggest mechanisms or their goals to overcome this limitation in potential future work.

Author Response

We kindly wish to thank you for your comments.

We hope that now we have fully addressed to all of them.

Our inputs in the main text highlighted in blue refers to the improvements addressed to your comments.